# Flexible Mechanoluminescent SrAl_2_O_4_:Eu Film with Tracking Performance of CFRP Fracture Phenomena

**DOI:** 10.3390/s22155476

**Published:** 2022-07-22

**Authors:** Yuki Fujio, Chao-Nan Xu, Nao Terasaki

**Affiliations:** 1Sensing System Research Center, National Institute of Advanced Industrial Science and Technology (AIST), Saga 841-0052, Japan; cn-xu@aist.go.jp (C.-N.X.); nao-terasaki@aist.go.jp (N.T.); 2Department of Molecular and Material Science, Kyushu University, Fukuoka 816-8580, Japan

**Keywords:** mechanoluminescence, SrAl_2_O_4_:Eu, fracture visualization, strain distribution

## Abstract

Non-destructive testing of carbon-fiber-reinforced plastic (CFRP) laminates with bidirectional fiber bundles (twill-weave) using a mechanoluminescence (ML) technique was proposed. The dynamic strain distributions and fracture phenomena of the CFRP laminates in the tensile testing were evaluated by the fabricated ML sensor consisting of SrAl_2_O_4_:Eu (SAOE) powder and epoxy resin. The ML images for the ML sensor attached to the CFRP laminates with bidirectional fiber bundles gave a net-like ML intensity distribution similar to the original twill weave pattern. Specifically, it was found that the ML intensity on the longitudinal fiber bundle, which is the same as the tensile direction, is higher than that on the transverse fiber bundle. This indicates that the ML sensor can visualize the load share between fiber bundles in different directions of the CFRP laminate with high spatial resolution. Meanwhile, the ML sensor could also visualize the ultrafast discontinuous fracture process of the CFRP laminates and its stress distribution. The amount of SAOE powder in the ML sensor affects the tracking performance of the crack propagation. A higher SAOE amount leads to a fracture of the ML sensor itself, and a lower SAOE amount leads to poor ML characteristics.

## 1. Introduction

CFRP is manufactured as laminated structures of differently oriented layers to meet the required anisotropic strength [1,2,3]. However, when CFRP laminates are used in products, such a complex and anisotropic structure complicates the prediction of stress–strain distributions. Furthermore, the earliest damage to CFRP laminates is a transverse crack in the resin, which causes serious damage such as fiber breakage and delamination. To ensure the reliability of CFRP as a structural material, it is important to understand the damage generation, progress process, and fracture mechanism. For the damage generation and its behavior, there have been some reports which are limited to the damage assessment before and after static stress analysis [4,5]. If the damage generation and its behavior can be visualized during the mechanical testing, it can contribute to enhancing the performance and ensuring the reliability of CFRP products. Therefore, an in situ evaluation technique, which can provide a dynamic stress–strain distribution and track fracture phenomena, has been strongly demanded.

So far, there have been a lot of reports regarding the non-destructive testing (NDT) of CFRP laminates or products, such as an electrical strain gage [6,7,8], a fiber Bragg grating (FBG) sensor [6,8,9], soft X-ray radiography [9,10], an eddy current scanning [11,12], a digital image correlation (DIC) technique [7,13,14], flash thermography [12,15,16], and ultrasonic testing [16,17]. Among various reports, we have reported that a mechanoluminescence (ML) technique using a flexible composite film composed of ML material and organic resin can be a useful technique for visualizing the dynamic equivalent strain distribution [18,19,20,21,22]. The ML is a luminescence phenomenon induced by mechanical actions such as compression, tension, friction, or torsion. When the ML occurs in the elastic region, we especially refer to it as elasticoluminescence [23,24]. One of the features of the ML technique as an NDT is the luminescence intensity with a proportional relationship to the strain/stress and their speeds in dynamic elastic deformation [25]. For this proportional relationship, we have proven that the ML intensity distribution overlapped with the results of finite element analysis (FEA) [20,26,27]. Another feature is the simple manufacturing method of flexible ML film. The flexible ML films are prepared by coating methods such as screen-printing technique and spray-painting method, indicating that they can be easily placed on various objects with complicated architectures [18,19,20,21,22,23,24,25,26,27,28,29,30,31,32,33]. By utilizing the captivating luminescence characteristics and simple manufacturing process, we believe that the ML technique is one of the possible candidates for the NDT of the CFRP laminates and products.

In the present study, therefore, we examined the strain distribution of the CFRP laminates with bidirectional fiber bundles (twill woven) in tensile testing by using the fabricated ML sensor consisting of a mixture of SrAl_2_O_4_:Eu (SAOE) powder and epoxy resin. The SAOE was selected due to the highest spectral sensitivity of the Si image sensor (commercial CMOS camera) and the highest ML intensity among various ML materials with visible luminescence. In addition, the SAOE is a safe and stable material to be used as a luminous indicator of a watch and as an emergency position indicator. After evaluating the stress–strain distribution, we visualized the crack propagation process for CFRP laminates by using the tailored flexible ML films. As a result, it was found that an ML film with an adjusted mixing ratio of SAOE powder and resin can be applied to the visualization and tracking of the crack propagation process without breaking the ML film. It is the first attempt to apply the ML technique to the visualization of dynamic strain distribution and strain changes associated with crack propagation in composite materials with complex and anisotropic structures. The application of such a new NDT technique is expected to greatly contribute to improving the mechanical performance and reliability of the design of next-generation composite materials.

## 2. Materials and Methods

An ML sensor was fabricated by a screen-printing method using a composite of SAOE powder and epoxy resin. The SAOE powder was synthesized via a high temperature solid-state reaction route. Commercial SrCO_3_, Eu_2_O_3_ (Kanto Chemical Co., Inc., Tokyo, Japan), and α-Al_2_O_3_ (Kojundo Chemical Laboratory Co., Ltd., Tokyo, Japan) powders were initially mixed in an agate mortar with ethanol solution and subsequently calcined at 800 °C for 1 h in atmospheric air. The mixture powder was pulverized and mixed with a small amount of H_3_BO_3_ (FUJIFILM Wako Pure Chemical Corporation, Osaka, Japan) used as a flux, and then sintered at 1350 °C for 4 h under a reducing atmosphere of the gas flow of 5%H_2_/Ar. The obtained powder was pulverized in an agate mortar and sieved with a stainless mesh with an aperture of 46 μm to prepare SAOE fine powder with a uniform particle size of several micrometers. The crystal structure and chemical composition were elucidated by X-ray diffraction analysis, resulting in the prepared SAOE powder was in a monoclinic SrAl_2_O_4_ phase (PDF No.: 01-074-0794 in ICDD), similar to the previous reports [18,19,23,25,34]. An ML paste consisting of a mixture of the SAOE powder and epoxy resin was directly coated on the surface of the CFRP laminate by screen-printing and cured to fabricate an ML sensor. The screen-printing method provides a homogeneous dispersion of SAOE particles and a uniform thickness of the ML sensor on the CFRP laminate. The homogeneous dispersion of SAOE particles was confirmed by observation of photoluminescence and afterglow distributions as well as fluorescent microscope images.

The CFRP laminate was used twill-weave CFRP laminates (dynalite201, 45 vol%CF/PA66, Tepex, Brilon, Germany), as shown in Figure 1a. The CFRP laminates with twill-weave were denoted by twill-CFRP. The physical dimensions of the twill-CFRP were 200 mm in length, 30 mm in width, 1 mm thick. The sensing characteristics of the ML sensors painted on the surface of the twill-CFRP were evaluated under tensile testing (Figure 1b,c). It is known that changes in the operating temperature of the tensile testing affect the hardness of the ML sensor and the trapped carriers of SAOE particles, resulting in changes in the ML sensor’s response. In this study, therefore, the tensile testing was performed in a dark environment at room temperature with little temperature change. A commercial strain gage (Kyowa Electronic Instruments Co., Ltd., Tokyo, Japan) was also applied on the backside surface of the CFRP laminates with a commercial adhesive. The tensile testing was performed with a material testing machine (MTS 810, MTS Systems Co., Eden Prairie, MN, USA). The ML images and longitudinal strains during the tensile testing were recorded using a CMOS camera and a uniaxial strain gage connected to a data logger, respectively. Both devices were connected to a personal computer to simultaneously obtain the data. The digitized data (ML images) recorded by the CMOS camera are in 16-bit format. For quantitative and reproducible measurements, the ML sensor was once irradiated by a blue light-emitting diode (LED) for 1 min and kept under dark conditions for 5 min. This ML image recording system has already been reported elsewhere [21,32,33,34]. Here, the mechanical characteristics of the CFRP laminate may change by coating the ML layer on the CFRP laminate. Before evaluation by ML technique, we investigated the effect of the coated ML layer on the mechanical characteristics of the CFRP laminate via strain measurements during the tensile testing utilizing both CFRP laminates with and without the ML layer. As a result, there was little difference in strain and strain rate between the CFRP laminates with and without the ML layer under the same experimental conditions.

## 3. Results and Discussion

The ML technique provides a luminescence distribution reflected in a dynamic strain distribution by just recording ML images using a digital camera [26,33]. This is the advantage of in situ NDT. The ML intensity distributions of the ML sensor (90 × 30 mm in size) painted on the twill-CFRP specimen were evaluated under the tensile testing. The applied tensile load and loading speed were set to 13 kN and 1 kN/s, respectively. Figure 2 shows representative ML images of the attached ML sensor painted on the surface of the twill-CFRP at the different applying loads (0–13 kN). As can be seen from Figure 2, the ML sensor exhibited a net-like luminescence pattern, and this pattern became clearer with the increase in the tensile load. There are roughly two kinds of ML intensity areas: higher (brighter) and lower (darker) ML intensity areas. For comparison, the ML intensity distribution for CFRP laminates consisting of only unidirectional (longitudinal) fiber bundles (Sankyo Seisakusyo Co., Ltd., Osaka, Japan, denoted by uni-CFRP) was also evaluated. As a result, the ML intensity distribution for the twill-CFRP was completely different from that for the uni-CFRP. The ML sensor on the uni-CFRP showed a uniform luminescence distribution and the ML intensity increased with an increase in the tensile load, as in an evaluation of a metal alloy plate specimen [33]. This indicates that the uni-CFRP is extended uniformly by the tensile load. In contrast, the net-like luminescence pattern of the twill-CFRP indicates a complicated stress–strain distribution originating from the different orientations of the fiber bundles and a multi-laminated layer.

The time course of the ML intensities in darker and brighter areas and the longitudinal strain for the twill-CFRP are depicted in Figure 3a. The ML intensities in darker and brighter areas were estimated by using the average of each of ROI-1 and ROI-2 of 8 × 8 pixels, respectively. The positions of ROI-1 and ROI-2 were shown in the inset of Figure 3a. The strain obtained from the strain gage varied monotonically in the examined strain range of 0–0.8%, indicating that twill-CFRP deformed uniformly in a macroscopic view without plastic deformation. However, it was unclear if the strain was measured in the longitudinal or transverse fiber bundles due to the larger size of strain gage than a fiber bundle area (gage length: 2 mm; a fiber bundle area: 2 × 4 mm). Both ML intensities in darker (ROI-1) and brighter (ROI-2) areas also increased with increasing the tensile load. Figure 3b shows the dependence of the ML intensities in ROI-1 and ROI-2 on the tensile load for the twill-CFRP specimen. The darker and brighter ML intensity areas were identified as transverse and longitudinal fiber bundles, respectively, as represented in the inset photograph of Figure 3b. The ML intensities in both areas coincide with each other in the initial tensile load region of approximately 0–2 kN. This implies that the fiber bundles in different directions were equivalently deformed by the tensile load. Subsequently, both ML intensities gave different propensity to an increase in tensile load, and the slope of the ML intensity curve for longitudinal fiber bundles is approximately twice compared to transverse fiber bundles. The slopes of the ML intensity curves for the longitudinal and transverse fiber bundle areas were calculated to be approximately 1.2 × 10^3^ and 0.65 × 10^3^, respectively, in the tensile load range of 3–12 kN. Here, the ML intensity is proportional to the time variation of strain energy (strain and strain rate) [25,35,36,37]. Considering this feature of ML material, we believe that the deformation speed of the longitudinal fiber bundle is about twice the deformation rate of the transverse fiber bundle in the higher tensile load range. Based on these results, we can summarize the ML characteristics against the elastic deformation in this study. The ML sensor gave a linear relationship in the wide strain range of 0.05–0.8%, a high spatial resolution of the ML image (170 μm/pixel) depending on the pixel resolution of the digital camera and repeatable measurements by excitation process before testing. Such a superior features could be, therefore, considered as one of useful techniques for NDT.

To investigate the details of the ML intensity distribution for the twill-CFRP laminates, the transverse profile of ML intensity under different tensile loads in the range of 0~13 kN were compared and presented in Figure 3c. The transverse line (A-A’ line) was also shown in the inset of Figure 3c. The ML intensities on the A-A’ line increased with the increasing tensile load. The positions of the high and low ML intensities are found to be completely in accordance with the fiber bundle directions (the upper part of Figure 3c); the ML intensity for the longitudinal fiber bundles was much higher than the ML intensity for the transverse fiber bundles. This difference in the ML intensity due to the fiber bundle direction shows the same tendency as the literature written by G. Anzelotti et al. [13]. Each of the ML intensities for fiber bundles in the transverse direction (ROI-1) and the longitudinal direction (ROI-2) shows almost the same values, even when they are different from the locations shown in Figure 3a. This presumption is supported by the results of Figure 3c, and the ML intensities on fiber bundles with the same direction increase at almost the same rate as the increase in the applied load. Thus, we can conclude that the ML intensity difference between fiber bundles with different directions (Figure 3a) shows the same tendency even in different areas. Interestingly, there is a small decrease in the ML intensity at the interface between the longitudinal fiber bundles. This may imply that there is only a resin (polyamide) with a different Young’s modulus, demonstrating that the ML sensor sensitively responds to such a difference in Young’s modulus with a high spatial resolution. Here, there is a variation in the ML intensity of the longitudinal or transverse in different sites, as shown in Figure 3c. We can consider several potential reasons why the variation in the ML intensity appeared at different sites of the fiber bundle in the same direction. Among them, it is caused by the heterogeneous structure of CFRP laminate or measurement noise. Since it has been reported that the composite material consisting of resin and fiber fabric shows different mechanical characteristics depending on the manufacturing and processing conditions [2,4,7], the CFRP laminate used in this study may also have local strain changes. Therefore, in the tensile testing of the twill-CFRP via the ML sensor, it is concluded that both the longitudinal and transverse fiber bundles in twill-CFRP share the tensile load equivalently in the smaller load range of 0~2 kN, and then the longitudinal fiber bundles were shared at about twice load in comparison with the transverse fiber bundles. Based on the above discussed results, the ML sensor can be a potential candidate for the in situ NDT, as well as an engineering design tool for actual CFRP products with complicated architectures.

Incidentally, composite materials such as CFRP suddenly decrease the structure’s strength induced by the impact of the object. Such an indentation by impact becomes a stress concentration in the structures and poses significant damage such as matrix cracks, fiber breakage, or delamination [38]. Therefore, it is essential to visualize the stress concentration and understand the fracture phenomenon of the CFRP laminates in order to design a strong structure. To evaluate and visualize the fracture process of CFRP laminates, we investigated the possibility of the ML material as the fracture process visualization and the effect of the ML sensor structure on the characteristics of fracture process visualization. The pastes mixed with SAOE powder and epoxy resin (Ep) in various weight ratios were printed directly on the twill-CFRP (164 mm in length; 15 mm in width; 1 mm thick) using the same screen-printing method as above. After printing the ML sensor, an artificial notch as a crack initiation was introduced in the lateral side of the twill-CFRP (Figure 4a). The physical dimensions of the artificial notch were 5 mm in length and 0.5 mm in width. The ML images during the fracture process of twill-CFRP by the tensile testing were recorded by using a high-speed CMOS camera (FASTCAM Mini AX100, Photron Limited, Tokyo, Japan) in a monochrome 16-bit format with a frame rate of 30,000 Hz. The tensile testing was carried out by displacement control at a displacement rate of 1 mm/min. The applied tensile load was confirmed to vary almost linearly, and the maximum load reached 2.1 kN, as shown in Figure 4b. A fracture phenomenon starting from an artificial notch was simulated in mechanical testing by fixing both ends of the bare CFRP laminate, as shown in Figure 4a. That is, the crack propagated through the CFRP laminate. At this time, the ML sensor was vertically divided without peeling off from the CFRP laminate surface as the crack propagated.

Figure 4c shows a bright-field image under the ultraviolet (UV) light irradiation for the twill-CFRP with the ML sensor assembled in the material testing machine. The carbon fiber bundle direction on the surface of twill-CFRP can be seen through the thinner thickness of the ML sensor (approximately 40 μm), even when the ML sensor is covered on the surface of the twill-CFRP. The representative ML images for the twill-CFRP with an ML sensor (SAO/Ep = 60/40) in the tensile testing at a displacement speed of 1 mm/min are depicted in Figure 4d. From careful observation of the ML images, it was found that there was a weak and tiny ML pattern that appeared at the tip of the artificial notch at the beginning of the tensile test, and the ML pattern was kept until about 4500 μs before the fracture. Subsequently, as shown in Figure 4d, a strong ML with a vertical distribution was observed away from the tip of the artificial notch 4466 μs before the fracture. The vertical ML distribution moved to the horizontal direction tracking the crack propagation. That is to say, by applying the ML technique to the visualization of the fracture process of twill-CFRP, the crack propagation process and the related stress distributions could be visualized even in a considerably short time of 4500 μs. Figure 4e depicts schematic views of the carbon fiber bundle direction and the ML distribution on the surface of the twill-CFRP in imitation of Figure 4d. It is noteworthy that the strong ML distribution is located mainly between the longitudinal carbon fiber bundles. It assumes that crack propagation is temporarily arrested by the carbon fiber bundle next to the fractured carbon fiber bundle. In summary, it was proven that an ML technique combined with a high-speed camera could visualize the ultrafast discontinuous fracture process of anisotropic composite materials. 

Next, we examined the effect of the ML sensor structure consisting of SAOE powder and epoxy resin on the visualization and tracking performances for the crack propagation of the fracture process in twill-CFRP. Here, it has been reported that the ML characteristics of the composite of ML powder and optical resin are influenced by the thickness of the ML film and the composition of the ML powder [25]. However, the relationship between the ML characteristics and the tracking performance for crack propagation remains unknown. Considering both structural factors, it is presumed that a thick ML film interferes not only with the crack propagation but also with the original characteristics of twill-CFRP, even though it contributes to the creation of a higher ML intensity [25,39]. Thus, paying attention to the mixing ratio between SAOE powder and epoxy resin in the ML sensor, we prepared and evaluated the ML sensor consisting of SAOE powder (SAO) and epoxy resin (Ep) in various weight ratios (SAO/Ep = 40/60, 50/50, 60/40, 70/30, 80/20, 90/10).

The obtained ML images for each ML sensor at the fracture of the twill-CFRP are summarized in Figure 5(a1–a6). Figure 5(b1–b4) depicts optical microscope images of the twill-CFRP surface with an ML sensor after fracture testing. The optical microscopic observation of the CFRP laminate surface after the fracture testing is performed to confirm the splitting of the ML sensor along the crack propagation route and the exfoliation of the ML sensor from the CFRP laminate surface. It is seen that the ML sensors themselves are broken when the SAOE amount exceeds 80 wt.%. Especially, the ML sensor with 90 wt.% SAOE powder broke as soon as the crack propagation in the twill-CFRP started without tracking the crack propagation. In Figure 5(b3,b4), there is a waffle pattern on the surface of the ML sensors. This is similar to the mesh pattern of the screen mask used in the screen-printing for ML sensor fabrication. High amounts of SAOE powder not only increase the viscosity of the ML paste but also form an inhomogeneous surface of an ML film. Therefore, it is considered that such an ML sensor is unsuitable for visualizing the strain distribution and crack propagation of composite materials because it cannot provide the stress distribution of the measuring objects. In addition, it is found that the ML sensor had peeled off and a bare CFRP surface was exposed after the fracture testing. The ML pattern in a place unrelated to crack propagation, as shown in Figure 5(a5,a6), seems to be due to the exfoliation of the ML sensor from the twill-CFRP laminate. On the contrary, the ML sensors with a lower amount of SAOE powder exhibit good tracking performance with the crack propagation without the fracture of the ML sensor itself, although there is a little noisy pattern. The optical microscopic images after the fracture testing showed no exfoliation of the ML sensor from the surface of the CFRP laminate when the amount of epoxy resin was more than 40 wt.%. Thus, in the crack propagation evaluation, it is necessary to control the mixing ratio of the SAOE powder and epoxy resin. Considering the fracture mechanism for the interface between the ML sensor and the twill-CFRP, it is speculated that the fracture of the ML sensor without tracking the crack propagation was caused by the decrease in the adhesive strength at the interface between the ML sensor and the CFRP laminate. An increase in the SAOE amount in the ML sensor, in other words, a decrease in the amount of epoxy resin, can be a sufficient factor to reduce the adhesive affinity with CFRP with a polyamide resin. Therefore, it can be concluded that when the adhesive strength at the interface between the ML sensor and the CFRP is sufficiently higher than the fracture strength of the ML sensor, it shows superior tracking performance for crack propagation.

## 4. Conclusions

The strain distributions and crack propagation for CFRP laminates with bidirectional fiber bundles in the tensile testing were evaluated by using the ML technique. The ML sensor visualized the strain distribution of CFRP laminates with bidirectional fiber bundles, i.e., we can experimentally obtain the strain distribution of CFRPs with complicated architectures. In addition, the ML sensor detected a tiny strain difference at the interface between the longitudinal fiber bundles in the elastic deformation region. From the evaluation of the tracking performance with the crack propagation, the ML sensor visualized crack propagation as well as the related stress distribution. The higher SAOE amount leads to fractured the ML sensor itself, and the lower SAOE amount leads to poor ML intensity distribution. Thus, it can be concluded that the optimal SAOE amount for the ML sensors for tracking the crack propagation is approximately 60% in the weight ratio.

## Figures and Tables

**Figure 1 sensors-22-05476-f001:**
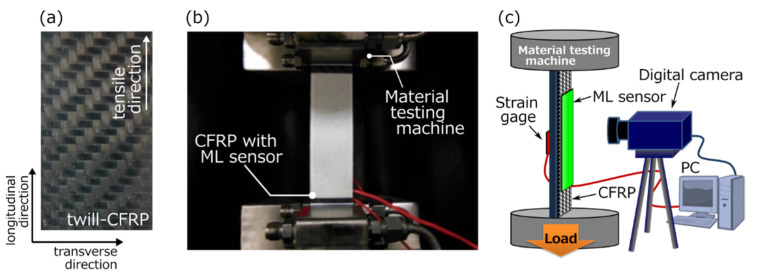
(**a**) Photographs of twill-CFRP with bidirectional fiber bundles, (**b**) twill-CFRP with ML sensor assembled in material testing machine, and (**c**) schematic illustration of experimental set up for the tensile testing.

**Figure 2 sensors-22-05476-f002:**
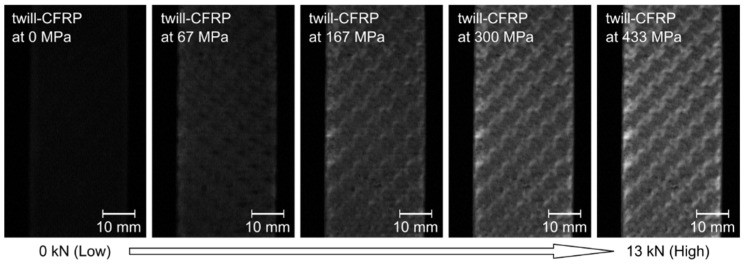
Representative ML images of the ML sensor attached to the twill-CFRP under the different applied tensile loads (0, 2, 5, 9, and 13 kN).

**Figure 3 sensors-22-05476-f003:**
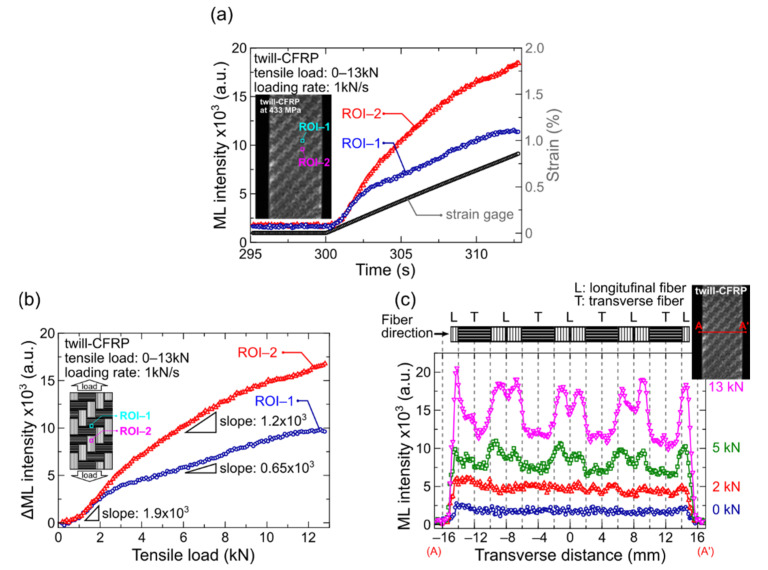
(**a**) ML sensor and strain gage responses to the tensile load applied to the twill-CFRP in the range of 0–13 kN (inset: positions of ROI–1 and ROI–2 on ML image), (**b**) dependence of ML intensity on the tensile load for twill-CFRP (inset: illustration of twill-CFRP with ROI–1 and –2 marks), and (**c**) ML intensity profiles along A–A’ transverse line in inset ML image under different tensile loads in the range of 0–13 kN.

**Figure 4 sensors-22-05476-f004:**
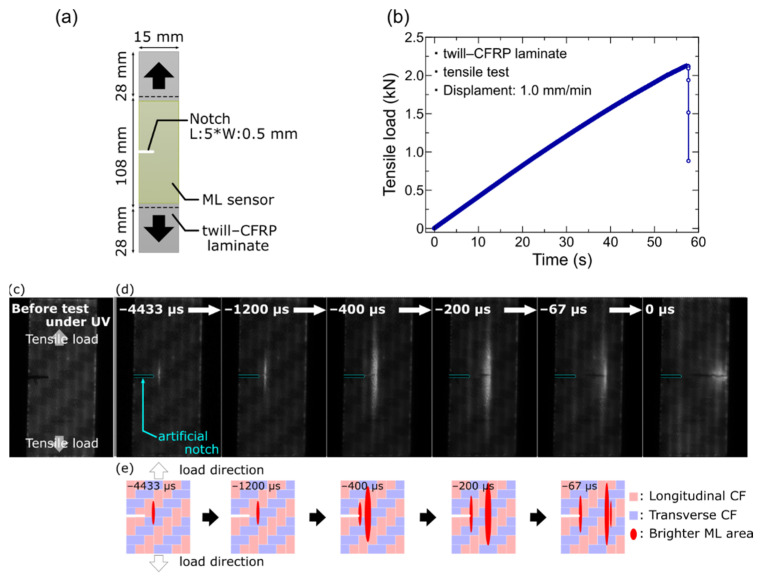
(**a**) Schematic illustration of twill-CFRP with artificial notch covered with ML sensor, (**b**) tensile load curve for twill CFRP with artificial notch under tensile testing at a displacement rate of 1mm/min. When the tensile load was approximately 2.1kN, the twill–CFRP was fractured. (**c**) Photograph of twill-CFRP with ML sensor set in the material testing machine under UV irradiation, (**d**) representative ML images of a crack propagation process in the tensile testing, and (**e**) schematic illustration of twill-CFRP surface with bright ML areas.

**Figure 5 sensors-22-05476-f005:**
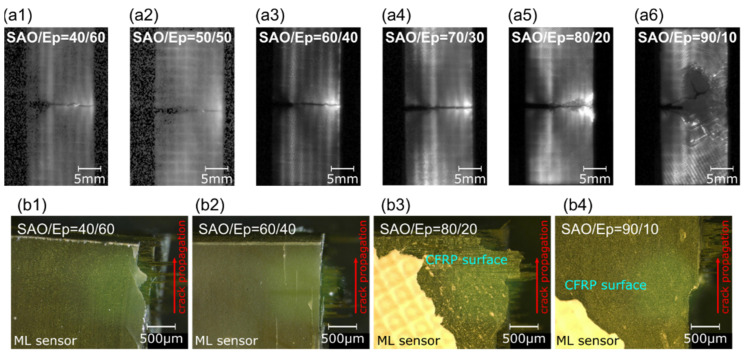
(**a1**–**a6**) ML images of twill-CFRP laminates attached with ML sensors composed of SrAl_2_O_4_ powder and epoxy resin in the various weight ratio and (**b1**–**b4**) optical microscope images for twill-CFRP laminate surface with ML sensor after fracture testing.

## Data Availability

Not applicable.

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
