# Peer review of "Flexible Mechanoluminescent SrAl2O4:Eu Film with Tracking Performance of CFRP Fracture Phenomena"

_sensors, 2022, doi:10.3390/s22155476_

Round 1
Reviewer 1 Report
In the reviewed work a method for non-destructive testing of carbon fiber reinforced plastic (CFRP) laminates by applying a mechanoluminescence (ML) technique was described. The ML sensor consisting of SrAl2O4:Eu (SAOE) powder and epoxy resin was used to cover CFRPs, and it was found that the ML sensor is able to visualize the load share between fiber bundles with different directions in CFRP laminate with high spatial resolution. It is an interesting approach for non-destructive testing of CFRP laminates, however, there are some points that need to addressed in more depth:
- „The ML paste consisting of a mixture of the SAOE powder and epoxy resin was directly coated on the surface of the CFRP laminates by the screen-printing, and it was cured to fabricate an ML sensor” – please verify and provide evidence that SAOE particles were homogeneously distributed on CFRP surface;
- what about the influence of temperature on ML sensor response?
- what are the sensitivity, resolution, and repeatability of the sensor?
- SAOE (SrAl2O4:Eu) – please provide information on satability and safety issues of this compound (toxicity, etc.).
Reviewer 2 Report
This paper is interesting , although I have some comments about the content and presentation made.
The originality of the work must be emphasized,
the study is correct although it could benefit if some points are emphasized, for example we can wonder is the ML intensity is really due to the composite strain or to the layer of sensor... would it be possible to correlate the strain field of the composite with another technique?
The figure 3 is very interesting.
The figure 4 is more questioning...the crack in the composite or in the sensor layer...
The figure 5 in my sense definitely needs more explanations, in particular the b images, obtained by optical microscope.
Reviewer 3 Report
In the manuscript titled “Flexible Mechanoluminescent SrAl2O4:Eu Film with Tracking Performance of CFRP Fracture Phenomena”, Fujio et al. examined the strain distribution of the CFRP laminates with bidirectional fiber bundles (twill woven) in the tensile testing using the fabricated ML sensor consisting of SrAl2O4:Eu powder and epoxy resin. The present study is interesting but need to address following points before publishing.
1- Typos are found in many places in the text.
2- It is not clear if the distribution of ROI-1 and ROI-2 is same on the different sites of the test samples. Please explain.
3- What is the variation in the ML intensity of the longitudinal or transverse in different sites of tested sample?
4- As the CFRP laminates coated with the fabricated ML, the strain and strain rate will be affected. Please explain.
